# Effect of bariatric and metabolic surgery on rheumatoid arthritis outcomes: A systematic review

**Saoussen Miladi**[1,2], **Yasmine Makhlouf**[1,2]*, **Hiba Boussaa**[1,2], **Leith Zakraoui**[1,2], **Kawther Ben Abdelghani**[1,2], **Alia Fazaa**[1,2], **Ahmed Laatar**[1,2]

1 Department of Rheumatology, Mongi Slim Hospital, La Marsa, Tunis, Tunisia, 2 University Tunis El Manar, Tunis, Tunisia

* Yasmine.mkhlouf@gmail.com

## Abstract

### Introduction

Obesity is a growing and debilitating epidemic worldwide that is associated with an increased inflammation. It is often linked to rheumatic diseases and may impact negatively their natural history. The use of bariatric and metabolic surgery (BMS) has increased thanks to its positive effect on major comorbidities like diabetes type 2. This systematic review provides the most up-to-date published literature regarding the effect of BMS on outcomes in rheumatoid arthritis.

### Methods

This systematic review followed the preferred reporting items for systematic reviews guidelines. Original articles from Pubmed, Embase and Cochrane, published until June 16th 2023, and tackling the effect of BMS on disease outcomes in patients with RA were included.

### Results

Three studies met the inclusion criteria. They were published between 2015 and 2022. The total number of RA patients was 33193 and 6700 of them underwent BMS. Compared to non-surgical patients, weight loss after BMS was associated with lower disease activity outcomes at 12 months (p<0.05). Similarly, prior BMS in RA patients was significantly associated with reduced odds ratios for all the morbidities and in-hospital mortality compared with no prior BMS (36.5% vs 54.6%, OR = 0.45, 95% CI (0.42, 0.48), p< 0.001) and (0.4% vs 0.9%, OR = 0.41, 95% CI (0.27–0.61), p < 0.001) respectively.

### Conclusion

To conclude, published data indicate that BMS seems a promising alternative in reducing RA disease activity as well as morbidity and mortality in patients with obesity.

**Data Availability Statement:** All relevant data are within the paper and its Supporting Information files.

**Funding:** The authors received no specific funding for this work.

**Competing interests:** The authors have declared that no competing interests exist.

# 1 Introduction

Obesity is a growing and debilitating epidemic worldwide. According to the World Health Organization, one-third of the population suffers from obesity or overweight [1]. Adipose tissue is considered as a highly dynamic organ that maintains normal metabolic function and energy homeostasis. Indeed, there is a close interaction between metabolism and immune system, and a higher body mass index (BMI) is associated with an increased inflammation [2]. More importantly, obesity is often linked to other diseases including rheumatic diseases and may impact negatively their natural history [3]. Particularly, higher levels of adiposity were associated with higher risk of developing rheumatoid arthritis (RA) (RR = 1.25, 95% CI[1.07–1.45], P <0.01) [4]. Similarly, RA patients suffering from obesity were less likely to achieve remission or low disease activity [5]. Indeed, adipocytokines produced by the adipose tissue maintain an inflammatory state in the synoviocytes which makes it difficult to achieve remission [6]. Thus, there is a need for more stringent interventions to reduce weight in this population.

The use of bariatric and metabolic surgery (BMS) has increased thanks to its positive effect on major comorbidities like diabetes type 2 [1]. It is hypothesized that weight loss is associated with a reduction of adipokine levels which improves outcomes in RA [4]. While non-surgical weight loss improved outcomes in RA, the effect of BMS in this context is not well defined [7].

For a more comprehensive assessment, we conducted this systematic review to provide the most up-to-date published literature regarding the effect of BMS on outcomes of RA including disease activity as well as RA morbidity and mortality.

# 2 Methods

This systematic review followed the preferred reporting items for systematic reviews guidelines. All data analyzed were extracted from published studies. For the present paper, no ethical approval or written informed consent was required. The search strategy, literature selection, and data extraction were conducted by two investigators (SM and YM) independently, then discussed, and any disagreement was resolved by consensus.

## 2.1 Search strategy

Eligible articles were searched in Medline, Embase, and Cochrane Library. For PubMed, the search was carried out using a strategy employing the combination of the MeSH (Medical Subject Headings) terms. The keywords used were "rheumatoid arthritis", "Arthritis", "Inflammatory disease", "Immune-mediated rheumatic disease", "Obesity management", "bariatric surgery", "Gastric Bypass", "Gastroplasty", "Patient Reported Outcome Measures", "Blood Sedimentation", "C-Reactive Protein", "mortality" and "morbidity". For Embase and Cochrane Library, the previous terms were searched in the article title, abstract, or keywords.

## 2.2 Selection criteria

A comprehensive search was conducted from inception until June 16th 2023. The inclusion criteria for the present systematic review were: 1) Patients who underwent a BMS. 2) Patients followed for RA with available data on disease outcomes before and after BMS. 3) Comparison of RA outcomes before and after surgery with or without a control group. 5) Cohort studies assessing RA outcomes over time.

Only original articles written in English were considered. Publications not in compliance with this systematic review purpose as well as those not representing original research (*i.e.*; meta-analyses, reviews, editorials, qualitative papers, case reports, comments, and letters to

editors) were excluded. Additional articles were manually retrieved based on the references of selected articles. If any study included overlapping data, the most comprehensive one was selected. After a deep analysis of titles and abstracts, articles unrelated to the inclusion criteria were excluded.

## 2.3 Data extraction and quality assessment

Extracted data from each study was evaluated independently by both investigators (SM and YM). A pilot-tested extraction form (Zotero) was used by both then compared between the two investigators. The extracted data included the main methodological characteristics of the articles: study data (year of publication, country, study design, number of included subjects, mean age of included subjects, inclusion and exclusion criteria, duration of the follow-up).

Our primary outcome was the evaluation of the effect of BMS on disease activity as well as on morbidity and mortality of RA patients. Furthermore, we identified potential biases of the cohort studies using the Newcastle—Ottawa Quality Assessment Scale. Only studies that met high quality belonged to our final selection.

## 3 Results

The initial search yielded 103 papers. Following duplicate elimination, we screened for 90 papers. Overall, 3 papers were finally selected for analysis. The flow chart of this systematic review is summarized in **Fig 1**.

### 3.1 Characteristics of the studies

The main characteristics of the 3 studies selected in this systematic review are represented in **S1 Table** [7–9]. Three studies published between 2015 [7] and 2022 [8] included RA patients who underwent BMS. The studies were conducted in China [9], Boston [7] and Taiwan [8]. Both were cohort studies [7, 9] and the other one a case control study [8]. The control group included patients with obesity who did not receive any intervention. The exclusion criteria were mentioned in only 2 studies. A history of malignancy or problems contraindicating BMS and secondary obesity were cited in the study of Fang et al. [9]. In the study of Sparks and Lin et al., only subjects who had adequate clinical data available before and after BMS were included in the study [7, 8].

### 3.2 Characteristics of the patients

Overall, 33193 patients were included and the sample sizes varied from 53 [7] to 33,075 [8]. The mean age was 52.1 years. Extremes were mentioned only in one study [8]. The mean number of RA patient who underwent BS was 6700 [32–6617] [8, 9]. The mean disease duration of RA before BS was 9 years with extremes ranging between 8.5 and 10.3 years [7, 9]. The diagnosis criteria applied in the studies were 1987 American College of Rheumatology RA [7], the Classification of Diseases (ICD) codes for RA [8] and the 2010 ACR/EULAR classification criteria [9].

All the selected patients were obese. Anthropometric data was available in two studies. The mean BMI at inclusion and after BS was 43.1 kg/m$^2$ and 29.7 kg/m$^2$ respectively [7, 9], with a mean loss change estimated at -34.2 (p = 0.01).

### 3.3 The effect of bariatric surgery on disease activity

**3.3.1 Inflammatory markers and disease activity outcomes.** The effect of BMS on disease activity was reported in two studies [7, 9]. Disease activity was evaluated at 4, 6, 8,12

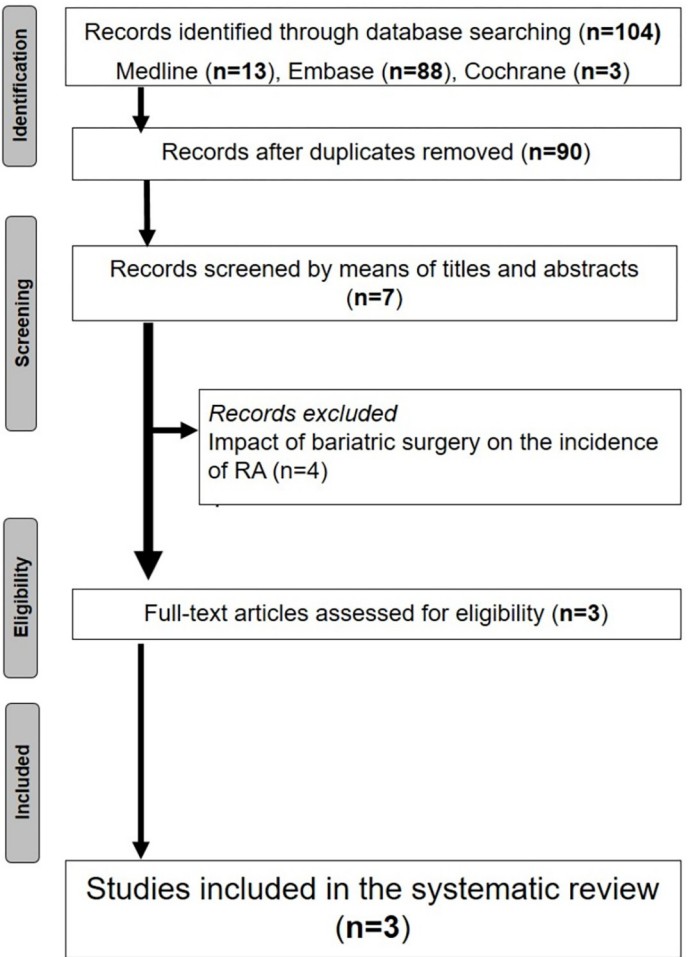

**Fig 1. Flow chart outlining the studied protocol.**

months all studies combined. The used scores were DAS28 ESR, DAS28 CRP, CDAI and ACR responses [9]. In the study by Sparks et al., validated measures were unavailable in most patients; assessment was based on an agreed upon a priori protocol [7]. Compared to non-surgical patients, weight loss after BMS was associated with lower disease activity. At 12 months post-surgery, 68% of subjects were in remission compared to 26% at baseline (p<0.001) [7]. An ACR20, 50 and 70 response rate were observed in 75.0%, 53.1% and 31.3% in the BMS compared to 51.5%, 39.4%, 21.2% in the non-surgery group (p < 0.01, p < 0.01, p < 0.01) respectively [9]. Regarding acute phase reactants, CRP and ESR were significantly lower at 12 months post-surgery (5.9 mg/L (SD 8.2), 26.1 mm/hr (SD 2.0)) compared to baseline (26.1 mg/L (SD 20.9), 45.7 mm/hr (SD 26.2)), (p<0.05, p<0.001) respectively [7]. In the study by Fang et al., the improvement in PhGA, EGP, CRP and ESR were not precised. The distribution of the different disease activity measures is represented in **Table 1**.

 **3.3.2 Medication tapering.** Medication tapering was reported in two studies [7, 9]. In the study by Fang et al., there was a significant reduction in the use of NSAIDs, Leflunomide, biologics and combination treatment in both surgery and non-surgery groups at 12 months compared to baseline (p < 0.01) (Table 2) [9]. This reduction did not concern corticosteroids, Methotrexate and Sulfasalazine use. Similarly, medication tapering in the bariatric group was

**Table 1. The distribution of disease activity measures at baseline and at follow-up before and after bariatric and metabolic surgery.**

| | | | Baseline | 4M | 6M | 8M | 12M | Most recent follow-up (mean 69.6 M) |
|---|---|---|---|---|---|---|---|---|
| Fang et al. (2020) | DAS28-ESR (S/NS) mean (SD) | | 6.3 (1.4)/ 6.2 (1.0)* | 3.1 (1.2)/4.1 (1.3)* | - | 2.0 (1.5)/ 2.8 (1.1)* | 1.5 (0.9)/ 2.4 (1.4)* | - |
| | DAS28-CRP (S/NS) mean (SD) | | 6.6 (3.7)/ 5.8 (2.9)* | 2.9 (2.1)/ 3.8 (1.2)* | - | 1.7 (1.4)/ 2.7 (1.8)* | 1.2 (0.9)/ 2.2 (1.7)* | - |
| | CDAI (S/NS) mean (SD) | | 38.9(24.5)/ 37.5(21.0)* | 21.4(18.3)/25.3 (17.5)* | - | 14.5(8.9)/20.3 (14.2)* | 9.5(6.8)/15.8 (12.5)* | - |
| | Weight, kg | | 111.8 (11.8) | 92.8 (19.3)** | - | 86.2 (23.7)** | 78.2 (25.6)** | - |
| | BMI, Kg/m$^2$ | | 38.4 (4.8) | 31.9 (10.2)** | - | 29.6 (11.9)** | 26.9 (13.5)** | - |
| Sparks et al. (2015) | RA disease activity (%)§ | Remission 26 | | - | 72** | - | 68** | 74** |
| | | Low 17 | | - | 23** | - | 17** | 23** |
| | | Moderate 51 | | - | 4** | - | 6** | 2** |
| | | High 6 | | - | 2** | - | 0** | 0** |
| | Weight, kg | | 128.2 (24.1) | | 95.8 (22.0)** | | 87.7 (20.0)** | 93.2 (24.0)** |
| | BMI, Kg/m$^2$ | | 47.8 (7.7) | | 35.7 (6.9)** | | 32.6 (7.0)** | 34.6 (8.0)** |

*p<0.05

**p<0.01, DAS28: 28-joint count disease activity score; ESR: erythrocyte sedimentation rate, CRP: C-reactive protein, CDAI: clinical disease activity index, S:surgery, NS: non-surgery, M = months, SD: standard deviation.

§ validated measures were unavailable in most patients, assessment was based on an agreed upon *a priori* protocol

not superior to that in non-surgical patients (p>0.05) [9]. In contrast, a significant decrease in the use of NSAIDs, glucocorticoids CsDMARDS and biologics was noted in the study by Sparks et al. [7]. Only one patient was in remission on no RA-related medications at baseline compared to 12 (23%), however this was not statistically significant (p = 0.28) at 12 months. Medication tapering in the different studies is represented in Table 2.

### 3.4 The effect of bariatric surgery on disease activity on morbidity and mortality

Only the study of Lin et al. assessed this particular aspect [8]. The multivariate analysis of this study showed that prior BMS in RA patients was significantly and independently associated

**Table 2. Characteristics of patients' medication at baseline and at 12 months post surgery.**

| | Fang et al. (M0(%)/M12(%)) | | Sparks (M0(%)/M12(%)) | |
|---|---|---|---|---|
| | Surgery | Non surgery | Surgery | Non surgery |
| **CSDMARDs** | 96.9/ 88.5 | 93.9/75.0 | 93/59** | - |
| **Methotrexate** | 87.5 /59.4 | 81.8/71.4 | - | - |
| **Sulfasalazine** | 15.6/9.4 | 12.1/7.1 | - | - |
| **Leflunomide** | 56.3/ 19.2** | 45.5/17.9* | - | - |
| **Combination** | 65.6/28.1* | 57.6 /28.6* | 70/43* | - |
| **Biological agents** | 43.8/15.6* | 30.3/21.4 | 51/36** | - |
| *Glucocrticoids* | 12.5/0 | 15.2/ 3.6 | 17/9* | - |
| *NSAIDs* | 50.0/6.3** | 51.5/25.0* | 45/15* | - |

*P < 0.05 and

**P < 0.01, csDMARDs: conventional disease-modifying anti-rheumatic drug, NSAIDs: non-steroidal anti-inflammatory drugs, M:months

with reduced odds ratios for all the morbidities compared with no prior BMS (OR = 36.5% vs 54.6%, OR = 0.45, 95% CI (0.42, 0.48), p< 0.001)). Similarly, this also concerned in-hospital mortality (0.4% vs 0.9%, OR = 0.41, 95% CI (0.27–0.61), p < 0.001)) [8].

## 4 Discussion

This systematic review investigated the impact of BS on clinical outcomes such as disease activity, morbidity and mortality in RA patients. The anti-inflammatory role was investigated in other diseases such as systemic lupus erythematosus and multiple sclerosis. Prior BMS was associated with an improvement in clinical course and in-hospitals outcomes compared to non-surgery [10, 11].

The two take-home messages derived from this systematic review are the following: i) Weight loss from BMS was associated with an improvement in disease activity outcomes in RA patients compared with no intervention. ii) RA patients with prior BMS were less likely to develop major morbidities and have a decreased in-hospital mortality compared to RA patients with obesity.

The achievement of an inactive disease state, or at least low disease activity, is the ultimate goal for patients with RA [12]. In addition to medication, non-pharmacological measures including weight loss are also at the core of the management of the disease. Previous studies have demonstrated that obesity decreases the likelihood of achieving a sustained remission despite adequate treatment [13–15]. Indeed, a recent meta-analysis revealed that odds of achieving a sustained remission were reduced by 51% when comparing RA patients with and without obesity [12]. However, results on whether obesity increase the incidence of RA are still conflicting and paradoxal [1, 5, 16–18]. This was particularly debated for the entity seronegative RA and obesity [1]. This may be explained by the marginal role of adipokines in the pathogenesis of RA as evidenced in the study by Qin et al. [4]. Surprisingly, BS seems to be a cause of developing RA.

Only few studies evaluated the role of non-surgical weight loss on RA outcomes [19, 20]. This review highlighted the important role of BMS on reducing serum inflammatory markers and disease activity. Beyond the weight loss itself, BMS seemed to reduce the inflammatory pathway. This was supported by the fact that the adipose tissue produce adipocytokines and inflammatory cytokines, which maintain an inflammatory state in the synoviocytes [6]. Another hypothesis included alterations in the microbiome and hormones, particularly glucagon-like peptide-1, which modulate inflammation in RA disease activity [21].

Similarly, current evidence seems to show improved outcomes in obese patients with other rheumatic disorders after bariatric surgery such as psoriatic arthritis, systemic lupus erythematosus (SLE) and gout [1, 22]. The positive effect was also was extended to a reduction in the use of corticosteroids as well as DMARDs. It is not clear whether the observed medication reduction is due to perioperative medication adjustments to avoid complications or to the bariatric surgery itself. However, 23% remained free of all RA medication one year after BMS in the study by Sparks et al. [7]. This contrasts with prior studies in which only 12% of RA patients remained free of medication two years after clinical remission without any BS [23].

As another highlight of our review, we focused on the effect of BMS on reducing morbidity and mortality of RA patients. The results of the study of Lin et al. including 6615 RA patients were unanimous [8]. BS could be a promising alternative in reducing comorbidities.

To the best of the authors' knowledge, this is the first systematic review to investigate the effect of BS on different aspects of RA including not only disease activity but also morbidity and mortality. Besides, our systematic review focused also on medication tapering after BS as part of disease activity evaluation.

However, some limitations should be addressed for this review. On the one hand, the number of the investigated studies was limited in the literature. More importantly, such interventions are not risk-free and are not performed as frequently as intended to conduct sufficient trials. Despite that, this review included an adequate number of patients to make statistically significant results and the quality of the studies was rated as high according to the NOS. On the other hand, diagnosis criteria and disease activity outcomes regarding RA varied according to authors. Consequently, comparison and interpretation of the results were difficult. Another potential limitation is the lack of a comparison group that utilized data collected in routine medical care, the non-randomized design as well as the short-term follow-up. Similarly, the doctors and patients were not blinded to the therapeutic strategy.

Finally, some factors such as dietary intake and physical activity were not collected, which might confound the efficacy of BS on disease activity. More importantly, these results should be interpreted with caution. Indeed, some authors showed that bariatric procedures were sometimes associated with an excessive weight loss and therefore an increased risk of death as well as a deleterious effect on bone [1, 24, 25].

Hopefully, future trials should tackle these particular issues. These studies should include more cases and control subjects to ascertain the specific implication of each subset for a better holistic approach.

### 4.1 Conclusion

Current evidence seems to show improved outcomes in RA patients with obesity one year after BMS. More rigorous prospective controlled studies with long follow-up are needed to ascertain the beneficial effect of such interventions.

## Supporting information

**S1 Checklist. PRISMA 2020 checklist.**
(DOCX)

**S1 Table. Main characteristics and results of the selected studies aiming to evaluate the outcomes of bariatric surgery on RA patients.**
(DOCX)

**S2 Table. Risk of bias assessment according to the Newcastle Ottawa Scale (NOS) for the cohort studies.**
(DOCX)

## Acknowledgments

Registration and protocol: The study is registered (PROSPERO) CRD42023437401

## Author Contributions

**Conceptualization:** Saoussen Miladi, Ahmed Laatar.

**Data curation:** Saoussen Miladi, Yasmine Makhlouf.

**Formal analysis:** Saoussen Miladi, Yasmine Makhlouf, Hiba Boussaa.

**Funding acquisition:** Kawther Ben Abdelghani.

**Investigation:** Saoussen Miladi, Yasmine Makhlouf, Hiba Boussaa, Kawther Ben Abdelghani.

**Methodology:** Saoussen Miladi, Kawther Ben Abdelghani.

**Project administration:** Kawther Ben Abdelghani, Alia Fazaa.

**Supervision:** Leith Zakraoui, Alia Fazaa.

**Validation:** Yasmine Makhlouf, Leith Zakraoui, Alia Fazaa, Ahmed Laatar.

**Visualization:** Leith Zakraoui, Alia Fazaa, Ahmed Laatar.

**Writing – original draft:** Yasmine Makhlouf.

**Writing – review & editing:** Yasmine Makhlouf.

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
