## [Decision Letter · Decision Letter 0]

16 Oct 2023

PONE-D-23-29166Effect of bariatric surgery on Rheumatoid Arthritis outcomes: A systematic reviewPLOS ONE

Dear Dr. Makhlouf,

Thank you for submitting your manuscript to PLOS ONE. After careful consideration, we feel that it has merit but does not fully meet PLOS ONE’s publication criteria as it currently stands. Therefore, we invite you to submit a revised version of the manuscript that addresses the points raised during the review process.

ACADEMIC EDITOR: Dear authors, thank you for submitting this manuscript to PLOS One. The reviewers have given comments regarding your manuscript, and I suggest you address each one.

We look forward to receiving your revised manuscript.

Kind regards,

Belal Nedal Sabbah

Academic Editor

PLOS ONE

Journal Requirements:

Reviewers' comments:

Reviewer's Responses to Questions

**Comments to the Author**

1. Is the manuscript technically sound, and do the data support the conclusions?

Reviewer #1: Yes

Reviewer #2: Partly

2. Has the statistical analysis been performed appropriately and rigorously? 

Reviewer #1: No

Reviewer #2: N/A

3. Have the authors made all data underlying the findings in their manuscript fully available?

Reviewer #1: No

Reviewer #2: No

4. Is the manuscript presented in an intelligible fashion and written in standard English?

Reviewer #1: Yes

Reviewer #2: Yes

5. Review Comments to the Author

Reviewer #1: Dear authors,

I read your manuscript carefully. I have few suggestions and questions and I hope addressing them, can increase the quality of your manuscript.

1- Please use the person first language and use "patients with obesity" instead of "obese patient".

2- Please clarify why you did the search for published papers between 2015 and 2022.

3- Please use "bariatric and metabolic surgery (BMS)" instead of bariatric surgery, alone.

4- Please add your searched "Keywords" to the "Search Strategy".

5- As different types of MBS have different effects and outcomes, I recommend to do a subgroup analysis between different types of MBS (sleeve gastrectomy, RYGB,.....)

6- There are other published systematic reviews on effect of BMS on other inflammatory/auto-immune diseases. I recommend to use these published papers at the beginning of discussion part to have a more broad vision on anti-inflammatory role of BMS.

- Impact of prior bariatric surgery on outcomes of hospitalized patients with systemic lupus erythematosus: a propensity score-matched analysis of the U.S. nationwide inpatient sample. (doi: 10.1016/j.soard.2023.06.006)

- Effect of Metabolic and Bariatric Surgery on the Clinical Course of Multiple Sclerosis in Patients with Severe Obesity: a Systematic Review. (DOI: 10.1007/s11695-023-06633-z)

Reviewer #2: Dear author,

This manuscript show clinical relevant. I did some comments to improve this manuscript:

Introduction:

It is recommended to provide more information about Rheumatoid Arthritis (RA) and how it relates to research outcomes. Clarify the research gap.

Methods:

It is necessary to inform the PROSPERO registration number

I suggest adding new MeSH (Medical Subject Headings) terms to the search, such as "Rheumatoid arthritis", "Arthritis", "Inflammatory disease", "Immune-mediated rheumatic disease", "Obesity management", "Gastric bypass", "Gastroplasty." Update your search with these new terms.

Specify which software (e.g. Mendeley, Zootero) was used for article and data extraction.

Results:

If one of the objectives was to examine the effect of bariatric surgery on disease activity, presentation of these data is suggested.

Include information about the methodological quality of the studies in the body of the text, as it is relevant to the discussion.

Consider creating a table with numerical clinical data (e.g., DAS-28, weight, BMI) before and after surgery to make it easier for readers to understand.

Discussion:

If the article claims that weight loss resulting from bariatric surgery is associated with improvements in RA outcomes, be sure to present this data clearly and accurately.

Dedicate a section to discussing the methodological quality of the included studies.

6. PLOS authors have the option to publish the peer review history of their article (what does this mean?). If published, this will include your full peer review and any attached files.

Reviewer #1: **Yes: **Mohammad Kermansaravi

Reviewer #2: No

---

## [Author Response · Author response to Decision Letter 0]

25 Oct 2023

Dear Editor, 

We would like to thank you for allowing us to resubmit a revised copy of this manuscript. We would also like to take this opportunity to express our thanks to the reviewers for the positive feedback and helpful comments for correction. We have revised the manuscript accordingly and provided a point-by-point response below. 

We hope the revised manuscript will be suitable for publication in your journal.

Best regards,

The corresponding author

1. Is the manuscript technically sound, and do the data support the conclusions?

Reviewer #1: Yes

Reviewer #2: Partly

Thank you for your answer

2. Has the statistical analysis been performed appropriately and rigorously?

Reviewer #1: No

Reviewer #2: N/A

Thank you for your answer. Indeed, we did not conduct a metanalyses.

3. Have the authors made all data underlying the findings in their manuscript fully available?

Reviewer #1: No

Reviewer #2: No

Thank you for your remark. We added the sentence in the declaration section: All the data is fully available without restriction.

4. Is the manuscript presented in an intelligible fashion and written in standard English?

Reviewer #1: Yes

Reviewer #2: Yes

Thank you for your remark.

5. Review Comments to the Author

Reviewer #1: 

1- Please use the person first language and use "patients with obesity" instead of "obese patient".

As requested, we replaced obese patients with patients with obesity (Line 67, Line 141, line 11 discussion, Line 66 conclusion)

2- Please clarify why you did the search for published papers between 2015 and 2022.

Thank you for your remark. A comprehensive search was conducted from inception until June 16th 2023. However, the included studies were published between 2015 and 2022 (Line 110).

3- Please use "bariatric and metabolic surgery (BMS)" instead of bariatric surgery, alone.

As requested, we replaced bariatric surgery with bariatric and metabolic surgery (BMS) in the abstract as well as in the manuscript (Line 49-Line 51-56�63,66, 87,88,93, 111, 112, 128, 139,143, 145, 160, 164, 167, 200, 201, discussion L4,L8,L9,L24, L25, L36, L39, L67).

4- Please add your searched "Keywords" to the "Search Strategy".

The keywords used were: “rheumatoid arthritis”, “bariatric surgery”, “Gastric Bypass”, “Gastroplasty”, “Patient Reported Outcome Measures”, "Blood Sedimentation", "C-Reactive Protein", “mortality” and “morbidity”. We added the keywords in the method section (LINE 104� LINE 107).

5- As different types of MBS have different effects and outcomes, I recommend to do a subgroup analysis between different types of MBS (sleeve gastrectomy, RYGB,.....)

Thank you for this pertinent remark. Indeed, it would be very interesting to compare the different interventions and outcomes in RA. Unfortunately, it was not possible to perform subgroup analyses in this particular matter as the different procedures were mentioned in the descriptive data and there were no statistical analyses performed to address its relation to patient reported outcomes. Moreover, the primary outcome did not focus on the type of intervention and the design of the study was not performed in that optic. However, this represents a great subject for future trials.

6- There are other published systematic reviews on effect of BMS on other inflammatory/auto-immune diseases. I recommend to use these published papers at the beginning of discussion part to have a more broad vision on anti-inflammatory role of BMS.

- Impact of prior bariatric surgery on outcomes of hospitalized patients with systemic lupus erythematosus: a propensity score-matched analysis of the U.S. nationwide inpatient sample. (doi: 10.1016/j.soard.2023.06.006)

- Effect of Metabolic and Bariatric Surgery on the Clinical Course of Multiple Sclerosis in Patients with Severe Obesity: a Systematic Review. (DOI: 10.1007/s11695-023-06633-z)

Thank you for your pertinent question. As requested, we highlighted the effect of BMS on author inflammatory/auto-immune diseases by adding these references at the beginning of the discussion: « The anti-inflammatory role was investigated in other diseases such as systemic lupus erythematosus and multiple sclerosis. Prior BMS was associated with an improvement in clinical course and in-hospitals outcomes compared to non-surgery (10,11) ». (Line 3-Line 6 discussion)

Reviewer #2: Dear author

1- Introduction:

It is recommended to provide more information about Rheumatoid Arthritis (RA) and how it relates to research outcomes. Clarify the research gap.

As requested, we clarified the research gap by emphasizing the fact that obesity is responsible for maintaining disease activity in RA patients. Moreover, data on the effect of surgical interventions on obesity and patient reported outcomes are lacking in RA.

Similarly, RA patients suffering from obesity were less likely to achieve remission or low disease activity (5). Indeed, adipocytokines produced by the adipose tissue maintain an inflammatory state in the synoviocytes which makes it difficult to achieve remission (6). Thus, there is a need for more stringent interventions to reduce weight in this population.(Line 82-86)

2- It is necessary to inform the PROSPERO registration number

As requested, we added the registration number at the bottom of the manuscript in declarations. The study is registered (PROSPERO) CRD42023437401.

3-I suggest adding new MeSH (Medical Subject Headings) terms to the search, such as "Rheumatoid arthritis", "Arthritis", "Inflammatory disease", "Immune-mediated rheumatic disease", "Obesity management", "Gastric bypass", "Gastroplasty." Update your search with these new terms.

4-Specify which software (e.g. Mendeley, Zootero) was used for article and data extraction.

As requested, we included the different Mesh words. The search was updated accordingly (Fig 1 flow chart).

We specified that we used Zotero for article and data extraction.

5-If one of the objectives was to examine the effect of bariatric surgery on disease activity, presentation of these data is suggested.

The evaluation of the effect of bariatric surgery on disease activity relied on acute phase reactants (ESR, CRP), disease activity scores: DAS28 ESR, DAS28 CRP, CDAI and ACR. We summarized data before and after surgey in paragraph 1.3.1 Inflammatory markers and disease activity outcomes and in Table 1.

6-Include information about the methodological quality of the studies in the body of the text, as it is relevant to the discussion.

The methodological quality of the studies was high according to Newcastle Ottawa Scale (NOS) scale which was added as a supplementary data. Moreover, we added more information and detail about the methodological qualities and limitations of the studies (Line 49-Line 55).

7-Consider creating a table with numerical clinical data (e.g., DAS-28, weight, BMI) before and after surgery to make it easier for readers to understand.

Regarding weight and BMI, they were added with disease activity in Table 1 as requested.

8-If the article claims that weight loss resulting from bariatric surgery is associated with improvements in RA outcomes, be sure to present this data clearly and accurately.

Dedicate a section to discussing the methodological quality of the included studies.

As requested, all the data regarding disease activity outcomes in RA patients was displayed in Table 1 and commented in the 1.3.1 Inflammatory markers and disease activity outcomes to facilitate the comprehension for the readers. 

A dedicated section discussing the methodological quality of the studies was added as requested (Line 46-Line 55).

---

## [Decision Letter · Decision Letter 1]

30 Oct 2023

Effect of bariatric surgery on Rheumatoid Arthritis outcomes: A systematic review

PONE-D-23-29166R1

Dear Dr. Makhlouf,

We’re pleased to inform you that your manuscript has been judged scientifically suitable for publication and will be formally accepted for publication once it meets all outstanding technical requirements.

Kind regards,

Belal Nedal Sabbah

Academic Editor

PLOS ONE

Additional Editor Comments (optional):

Reviewers' comments:

Reviewer's Responses to Questions

**Comments to the Author**

1. If the authors have adequately addressed your comments raised in a previous round of review and you feel that this manuscript is now acceptable for publication, you may indicate that here to bypass the “Comments to the Author” section, enter your conflict of interest statement in the “Confidential to Editor” section, and submit your "Accept" recommendation.

Reviewer #1: All comments have been addressed

Reviewer #2: All comments have been addressed

2. Is the manuscript technically sound, and do the data support the conclusions?

Reviewer #1: Yes

Reviewer #2: Yes

3. Has the statistical analysis been performed appropriately and rigorously? 

Reviewer #1: Yes

Reviewer #2: N/A

4. Have the authors made all data underlying the findings in their manuscript fully available?

Reviewer #1: Yes

Reviewer #2: Yes

5. Is the manuscript presented in an intelligible fashion and written in standard English?

Reviewer #1: Yes

Reviewer #2: Yes

6. Review Comments to the Author

Reviewer #1: Dear authors,

I read your revised manuscript and your responses.

I found all my comments addressed. Thank you for your revised manuscript.

Reviewer #2: (No Response)

7. PLOS authors have the option to publish the peer review history of their article (what does this mean?). If published, this will include your full peer review and any attached files.

Reviewer #1: **Yes: **Mohammad Kermansaravi

Reviewer #2: No

---

## [Editor Report · Acceptance letter]

7 Nov 2023

PONE-D-23-29166R1 

Effect of bariatric and metabolic surgery on Rheumatoid Arthritis outcomes: A systematic review 

Dear Dr. Makhlouf:

I'm pleased to inform you that your manuscript has been deemed suitable for publication in PLOS ONE. Congratulations! Your manuscript is now with our production department. 

Kind regards, 

on behalf of

Dr. Belal Nedal Sabbah 

Academic Editor

PLOS ONE